# Pediatric Asthma Attack and Home Paint Exposure

**DOI:** 10.3390/ijerph18084118

**Published:** 2021-04-13

**Authors:** Nadia T. Saif, Julia M. Janecki, Adam Wanner, Andrew A. Colin, Naresh Kumar

**Affiliations:** 1Department of Public Health Sciences, University of Miami Miller School of Medicine, Miami, FL 33136, USA; nadia.thura@gmail.com (N.T.S.); j.janecki@umiami.edu (J.M.J.); 2Division of Pulmonary and Sleep Medicine, University of Miami Health System, Miami, FL 33136, USA; AWanner@med.miami.edu; 3Division of Pediatric Pulmonology, Miller School of Medicine, University of Miami Health System, Miami, FL 33136, USA; acolin@med.miami.edu

**Keywords:** pediatric asthma, VOCs, paint exposure, home environment, allergies

## Abstract

Although asthma mortality has been declining for the past several decades, asthma morbidity is on the rise, largely due to deteriorating indoor air quality and comorbidities, such as allergies. Consumer products and building materials including paints emit volatile organic compounds (VOCs), such as propylene glycol (PG), which is shown to dehydrate respiratory tracts and can contributor to airway remodeling. We hypothesize that paint exposure increases the risk of asthma attacks among children because high levels of VOCs persist indoors for many weeks after painting. Children 1–15 years old visiting two of the University of Miami general pediatric clinics were screened for their history of asthma and paint exposure by interviewing their parents and/or guardians accompanying them to the clinic. They were also asked questions about asthma diagnosis, severity of asthma and allergies and their sociodemographics. The risk of asthma attack among asthmatic children was modeled with respect to paint exposure adjusting for potential confounders using multivariate logistic regressions. Of 163 children, 36 (22%) reported physician-diagnosed asthma and of these, 13 (33%) had an asthma attack during the last one year. Paint exposure was marginally significant in the univariate analysis (OR = 4.04; 95% CI = 0.90–18.87; *p* < 0.1). However, exposed asthmatic children were 10 times more likely to experience an asthma attack than unexposed asthmatic children (OR = 10.49; CI = 1.16–94.85, *p* < 0.05) when adjusted for other risk factors. Given paint is one of the sources of indoor VOCs, multiple strategies are warranted to manage the health effects of VOC exposure from paint, including the use of zero-VOC water-based paint, exposure avoidance and clinical interventions.

## 1. Introduction

Pediatric asthma remains one of the most common chronic diseases in the U.S., affecting 6.2 million children and accounting for approximately $50 billion per year in healthcare costs [1]. Although there has been a steady decline in asthma mortality, asthma and related allergic morbidity among children is on the rise [2]. Atopic (or allergy-induced) asthma accounts for over 75% of asthma cases [3,4]. Extraneous factors, including allergens (e.g., mold and pollen spores and endotoxins) and other environmental exposures, are the main drivers of atopic asthma [5]. Indoor pollutants emitted from consumer products and building material play an important role in the onset and persistence of asthma and allergies [6]. While the effects of indoor pollutants and allergens on asthma severity have been subject to research scrutiny [1,2], the role of paint, an importance source of volatile organic compounds (VOCs), is not well understood. For example, in a review of 20 studies found that, only one study demonstrated a significant association between domestic paint exposure and onset of asthma and asthma-like symptoms [7].

VOC exposure from paint: VOCs released from the dry paint have a complex chemical composition and certain compounds may exacerbate symptoms by irritating the heightened inflammatory response in the airways of asthmatics. One study found that low-VOC household products emit propylene glycol and glycol ethers (PG), which are associated with elevated risks of developing asthma, rhinitis and eczema in children [8]. PG emitted from low-VOC paint as it dries, warrants particular interest in understanding the mechanism by which VOCs from paint cause asthma attacks and exacerbation of symptoms.

The effects of acute and chronic inhalation of PG are becoming known as the molecule is more extensively studied due to its presence in e-cigarettes [9]. PG is a hygroscopic and when inhaled, it dehydrates the nasal, tracheal and respiratory tracts [9]. Chronic PG exposure even as low as 51 ppm is shown to cause nasal hemorrhaging and ocular discharge in rats, likely due to dehydration and irritation of the nasal cavity and eyes [9,10,11]. At medium and high doses of PG, respiratory epithelium thickens as a result of goblet cell metaplasia and increases mucin content [10,11]. Chronic dehydration and increased mucin production further decrease mucociliary clearance resulting in coughing and wheezing [12]. While the exact immunological mechanism induced by PG is still being investigated, the airway remodeling and resulting symptoms are similar to asthmatics’ response to exercise-induced bronchoconstriction (EIB) [13]. Based on this assumption, we propose airway remodeling mechanism in response to PG exposure as shown in Figure 1 [14]. In this study, we hypothesize that paint exposure increases the risk of asthma attack, especially among children with the preexisting risk factors, including allergies and exposure to other risk factors.

## 2. Methods and Material

### 2.1. Study Design

An observational cross-sectional design was used for screening children with asthma exacerbation and home paint exposure from two general pediatric clinics of the University of Miami Health System from June to October 2016. Children’s parents or guardians accompanying them to the clinics were consented and interviewed. The structured interview included questions on their children’s history of asthma, allergies, other comorbidities, medication use, paint exposure during the past year prior to visit the clinic, home environmental conditions, such as presence of smokers and pets in the house, and other sociodemographic characteristics. They were offered to be interviewed in the English or Spanish language. Their responses were recorded on an iPad using an automated data collection system designed Qualtrics. The study was approved by the University of Miami Institutional Review Board (IRB #20160040). Of 233 subjects contacted, 163 (70% response rate) consented and 157 completed the online survey. 

### 2.2. Outcome Measures

Our outcome measure was asthma attack, referred to as “asthma exacerbation” in this study. In addition, they were also screened for different allergy measures, including (1) allergic rhinitis, (2) eczema, (3) rhinitis symptoms (ever had sneezing, runny or blocked nose in the absence of a cold or flu; asked of all children whether or not they have a history of physician-diagnosed allergic rhinitis), (4) antiallergy medication use in the past month (as a proxy for allergy symptoms not otherwise diagnosed or specified by the other survey questions) and (5) testing positive for any allergen based on environmental allergy testing in the past. These questions were drawn from National Health and Nutrition Examination Survey (NHANES) 2005–2006 asthma and allergy content questions, the Behavioral Risk Factor Surveillance System (BRFSS) Asthma Call-Back Survey 2014 Child Questionnaire and the International Study of Asthma and Allergies in Childhood: Phase Three Core Questionnaire [15,16,17].

Question about paint included “During the last 12 months, were any areas inside your home painted, such as walls, trim or ceilings?”. Most paints even water-based paints have volatile organic compounds (VOCs) and half-life of organic VOCs is about 14 days, paint emits VOCs for many weeks after painting. Ideally, measurement of total VOCs and paint specific VOCs for many weeks after painting should be used to compute time-lagged exposure. However, acquiring such data was beyond the scope of this observation study. Therefore, response to the above question was used to develop proxy measure of exposure. Children whose home was painted during the past one year were classified as “exposed” and others as “unexposed” (see Appendix A for methodological details and the instrument).

### 2.3. Statistical Analysis

Analysis of data was conducted using STATA 14.0 [18]. A multivariate logistic regression model was employed to model the risk of asthma attack with respect to paint exposure adjusting for other risk factors, including smoking and allergy status.

## 3. Results

Of the 163 children included in the study, 36 (22%) had physician-diagnosed asthma and of these, 13 (36%) reported having an asthma attack (or exacerbation) within the last year, and 12 (33%) reported rhinitis symptoms during the past one year. Descriptive statistics and risk of asthma attack with respect to sociodemographic variables are presented in Appendix A.

Seven of the 26 children who were diagnosed with asthma and not exposed to paint, reported an asthma attack within the past year. Four of the 16 boys (0.25) and 3 of the 10 girls (0.30) had an asthma attack within the past year. Of the 10 children who were diagnosed with asthma and also exposed to paint, 2 of 5 boys (0.40) and 4 of 5 girls (0.80) reported an asthma attack within the last year. Thus, in both boys and girls who were diagnosed with asthma, those children who were exposed to paint also had a higher proportion of having an asthma attack (see Appendix A). In the bivariate analysis, paint exposure showed a marginally significant association with asthma attack (odds ratio (OR) = 4.04; 95% confidence interval (CI) = 0.90–18.87; *p* < 0.1) However, when adjusted for other risk factors, the association between asthma exposure and paint exposure strengthened. For example, independent association of secondhand smoke with asthma attack was insignificant. However, after adjusting for smoking, asthmatic children exposed to paint were 6.8 times more likely to have an asthma attack than unexposed children (see Appendix A). In the full model, the association of paint exposure almost doubled (OR = 10.49; 95% CI = 1.16–94.85; *p* < 0.05) adjusting for gender, age and other risk factors including allergies and medicine use for asthma and allergy management (Table 1).

## 4. Discussion

Our results show that the paint exposure is a significant risk factor of asthma attack while other environmental exposures further strengthen the association between paint exposure and asthma attack. Paints that are recommended for domestic hygiene are ubiquitously used. However, drying paint emits VOCs for many weeks to months after painting, resulting in the persistence of high levels of VOCs [19,20].

Emerging literature document that early exposure to indoor VOCs from renovation, furniture and other sources are associated with onset and persistence of asthma symptoms and symptom severity [21,22,23,24]. While respiratory health effects of VOCs have been subject to research scrutiny [6,25,26], this research makes a novel contribution by assessing the association of paint exposure with asthma attack among children. The findings of this research have health and policy implications for managing VOC exposure from paints.

### 4.1. Management Strategies and Policy Implications

Management strategies need to target sources, concentration reduction, exposure avoidance and clinical interventions. With regard to sources, switching to safer paint alternatives, such as low- or zero-VOC water-based paints, can reduce and eliminate VOC emission, because conventional oil-based paints emit high levels of VOCs and can result in chronic exposure to compounds such as PGs [11]. Still, caution is warranted because while the base-paint may be “zero-VOC”, adding color can increase VOCs as color dye contains VOCs [27,28]. Even low-VOC paint are associated with elevated risks of developing asthma, rhinitis and eczema in children [8]. Ventilation for many weeks after painting and use of a high efficiency air purifier equipped with activated carbon filter can reduce VOCs concentration as well [29]. Ideally, children, especially those with allergies and asthma, should not spend time in a painted home for about two to four weeks, given the 14 day average half-life of VOCs. Finally, pediatricians should provide information to the parents/guardians of children with asthma and allergies on potential ways to reduce paint exposure (see SOM for paint flyer). Additionally, they should review medicinal therapies with parents/guardians in anticipation of allergy and asthma symptom exacerbation in children who will likely be exposed to paint.

Besides paints, many consumer products founds in homes, including wood preservatives, aerosol sprays, cleaners/disinfectants, adhesives, air fresheners, printers and furniture fabrics, release VOCs [30]. According to the EPA, in the United States, the concentration of organic compounds is 2–5 times higher inside homes than outside [30]. As more research is conducted regarding the inhalation of VOCs such as PG and their pathophysiological impact, additional measures should be implemented to avoid predicted exposure to paint in all pediatric patients in order to avoid unknown health consequences. Among other measures, VOC reduction is warranted to manage an increasing burden of allergy and asthma morbidity.

### 4.2. Limitations

The findings of this research should be interpreted in light of its limitations. First, there is a possibility of selection bias in the sample because we collected data from two general pediatric clinics and only interested parents who had the time and literacy completed the survey. Notwithstanding, the survey was offered in both English and Spanish due to the diverse population of Miami. Second, the self-report survey by parents may be subject to recall bias, which may affect the results of age at diagnosis, asthma symptoms and time when the home was painted. Third, questions about paint lacked detailed information on paint type, paint color and time of painting, which can greatly influence the levels and persistence of VOCs and may violate, temporality, one of Hill’s criteria for causality. Fourth, our sample size was small to draw robust inferences. Thus, further research is warranted to replicate our findings of the epidemiological association between paint exposure and asthma attack in pediatric population. Finally, data were not collected on ventilation after painting, which can also affect both concentration and persistence of VOCs. Further research is warranted to address the above limitations including research on the assessment of time-lagged concentration of VOCs by different paint types, other sources of VOCs in home, use of air purifier and ventilation. This can help tease out the relative contribution of VOCs from different paints vis-à-vis VOCs from other consumer products adjusting for the effects of ventilation and air purifier. Such research can also guide evidence-based policy on domestic paint types, educational and clinical interventions and best practices to manage VOCs exposure and its associated health risks.

## 5. Conclusions

This study provides insight into the risk of paint exposure in a pediatric asthma attack. However, further research is warranted to develop empirical evidence of the efficacy of safer paints on the levels and persistence of VOCs after painting and VOCs from consumer products. In the meantime, educational and clinical interventions must be developed that can guide parents and policy makers in managing VOC exposure and its associated health risks.

## Figures and Tables

**Figure 1 ijerph-18-04118-f001:**
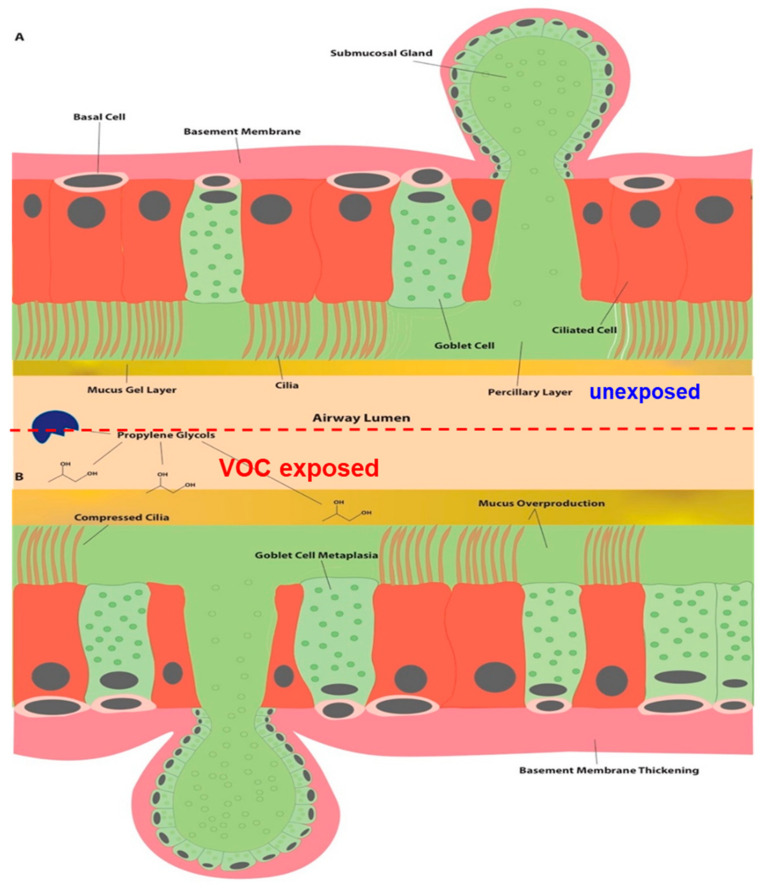
Airway remodeling resulting from inhalation of propylene glycol (PG) exposure. Panel (**A**) demonstrates a normal respiratory epithelium and periciliary layer. Panel (**B**) shows the proposed effect of chronic inhalation of PG. PG is hygroscopic and thus dehydrates the respiratory mucus gel layer, leading to an overproduction of mucus due to goblet cell metaplasia. The basement membrane also thickens as a result of dehydration. The thickened mucus layer also results in the compression of cilia causing decreased mucociliary clearance and exacerbation of asthma related symptoms such as coughing and wheezing.

**Table 1 ijerph-18-04118-t001:** Paint exposure and risk of asthma attack among children in Miami, FL, 2016.

Predictor	Model 4	
	OR	95% CI
Home paint during last one year ^a^	10.49 **	(1.161–94.851)
Smoker at home ^a^	0.16	(0.007–3.357)
Use of antihistamines for asthma and allergy management	1.11	(0.188–6.512)
Medicine use for asthma and allergy management ^a^	0.61	(0.043–8.676)
Age (year)	0.87	(0.678–1.103)
Gender (1 = boys, 2 = girls)	1.876	(0.368–9.557)

Abbreviations: OR, odds ratio; CI, confidence interval; ^a^ Variable was categorized as 1 = yes, 0 = otherwise. ** *p* < 0.05.

## Data Availability

The data used in this research are confidential and cannot be shared outside the study team. However, deidentified may be made available upon a reasonable request.

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
