# Peer review of "Pediatric Asthma Attack and Home Paint Exposure"

_ijerph, 2021, doi:10.3390/ijerph18084118_

Round 1

Reviewer 1 Report

The work "Pediatric Asthma Attack and Home Paint Exposure" is basically fine and the topic and also the title are very attractive. Several main concerns are as follows:

  1. The main problem of the work is that the sample (163 children) is relatively small. And therefore, the OR may be not reliable and the 95%CI range (lower to upper limit) is too large, as seen in Table 1 that the OR(95%CI) due to home paint during last one year was 10.49 (1.161- 94.851). I strongly suggest the authors discuss this in LIMITATIONS.
  2. TABLE 1. I suggest the authors to give the meanings of "**".
  3. CONCLUSIONS. I cannot fully agree to this statement "This study provides the first ever insight into the risk of paint exposure in pediatric asthma attack", this work may be not the first one (Please see the next comment).
  4. Several recent references indicated that (early life) exposure to house renovation or new furniture (that contribute to high level of indoor VOCs) during pregnancy or during the first several years after birth may be associated with childhood asthma, please refer to: (1) Deng et al. Effects of early life exposure to outdoor air pollution and indoor renovation on childhood asthma in China. Building and Environment 2015, 93: 84-91. (2) Rumchev et al. Association of domestic exposure to volatile organic compounds with asthma in young children. Thorax 2004;59:746–751. (3) Diez et al. Effects of indoor painting and smoking on airway symptoms in atopy risk children in the first year of life results of the LARS-study. Leipzig Allergy High-Risk Children Study Int. J. Hyg. Environ. Health, 203 (2000), pp. 23-28. (4) Franck et al. Prenatal VOC exposure and redecoration are related to wheezing in early infancy Environ. Int., 73 (2014), pp. 393-401. (5) Deng et al. Preconceptional, prenatal and postnatal exposure to outdoor and indoor environmental factors on allergic diseases/symptoms in preschool children. Chemosphere 2016, 152: 459-467.

Author Response

We are pleased to submit our revision and this PBP response for your consideration. We sincerely hope that you and reviewers find our revision satisfactory. Reviewers’ original comments are pasted as is followed by our responses in red color that begin with AR:

We are pleased to submit our revision and this PBP response for your consideration. We sincerely hope that you and reviewers find our revision satisfactory. Reviewers’ original comments are pasted as is followed by our responses in red color that begin with AR:

We are pleased to submit our revision and this PBP response for your consideration. We sincerely hope that you and reviewers find our revision satisfactory. Reviewers’ original comments are pasted as is followed by our responses in red color that begin with AR:

Reviewer 1

The work "Pediatric Asthma Attack and Home Paint Exposure" is basically fine and the topic and also the title are very attractive. Several main concerns are as follows:

AR: Thank you.

The main problem of the work is that the sample (163 children) is relatively small. And therefore, the OR may be not reliable and the 95%CI range (lower to upper limit) is too large, as seen in Table 1 that the OR(95%CI) due to home paint during last one year was 10.49 (1.161- 94.851). I strongly suggest the authors discuss this in LIMITATIONS.

AR: We fully agree. This limitation is document in the discussion

TABLE 1. I suggest the authors to give the meanings of "**".

AR: Described after Table 1.

CONCLUSIONS. I cannot fully agree to this statement "This study provides the first ever insight into the risk of paint exposure in pediatric asthma attack", this work may be not the first one (Please see the next comment).

AR: This statement is modified. Very true there are lots of research on VOCs emerging in recent years, and some of VOCs do come from paints (see line 200).

Several recent references indicated that (early life) exposure to house renovation or new furniture (that contribute to high level of indoor VOCs) during pregnancy or during the first several years after birth may be associated with childhood asthma, please refer to: (1) Deng et al. Effects of early life exposure to outdoor air pollution and indoor renovation on childhood asthma in China. Building and Environment 2015, 93: 84-91. (2) Rumchev et al. Association of domestic exposure to volatile organic compounds with asthma in young children. Thorax 2004;59:746–751. (3) Diez et al. Effects of indoor painting and smoking on airway symptoms in atopy risk children in the first year of life results of the LARS-study. Leipzig Allergy High-Risk Children Study Int. J. Hyg. Environ. Health, 203 (2000), pp. 23-28. (4) Franck et al. Prenatal VOC exposure and redecoration are related to wheezing in early infancy Environ. Int., 73 (2014), pp. 393-401. (5) Deng et al. Preconceptional, prenatal and postnatal exposure to outdoor and indoor environmental factors on allergic diseases/symptoms in preschool children. Chemosphere 2016, 152: 459-467.

AR: These references are cited in the paper (See line 150-52).

Reviewer 2 Report

As a reviewer I have the following remarks.

  1. Line 47. Full spell of VOC in the first use “source of VOCs”.
  2. Line 62: “At medium and high doses of PG” – doses or concentrations?
  3. Line 126: “7 of the 26 children who were diagnosed with asthma and not exposed to paint, re ported an asthma attack within the past year.” – is it possible that other factors are involved, say outdoor air pollution?
  4. Line 13. OR and CI are not fully spelled in the first use.
  5. Table 1. Gender or sex?
  6. Line 151. “the association of paint exposure with asthma attack.” – a similar association holds for adults?
  7. In Author Contribution some Dr need a dot (Dr.).
  8. The style of References is not homogenous. Adjust accordingly to the IJERPH’s specification (minor, say 21—23 vs. 24; et al. after one name or more?)
  9. In general, any information on paints- producer, variations, cheap vs. expensive etc.

Thank you

Author Response

Thank you for taking time to read and review our draft. But it looks like you provided us comments of someone else's manuscript on Malaria. 

Round 2

Reviewer 1 Report

The work is attractive.